# Cooperative Multiband Spectrum Sensing Using Radio Environment Maps and Neural Networks

**DOI:** 10.3390/s23115209

**Published:** 2023-05-30

**Authors:** Yanqueleth Molina-Tenorio, Alfonso Prieto-Guerrero, Rafael Aguilar-Gonzalez, Miguel Lopez-Benitez

**Affiliations:** 1Information Science and Technology Ph.D., Metropolitan Autonomous University, Mexico City 09360, Mexico; yanqueleth@xanum.uam.mx; 2Electrical Engineering Department, Metropolitan Autonomous University, Mexico City 09360, Mexico; apg@xanum.uam.mx; 3Faculty of Science, Autonomous University of San Luis Potosi, San Luis Potosi 78210, Mexico; 4Engineering Department, Arkansas State University Campus Queretaro, Queretaro 76270, Mexico; 5Department of Electrical Engineering and Electronics, University of Liverpool, Liverpool L69 3GJ, UK; mlopben@liverpool.ac.uk; 6ARIES Research Centre, Antonio de Nebrija University, 28040 Madrid, Spain

**Keywords:** multiband spectrum sensing, cognitive radios, radio environment maps, neural networks, cooperative sensor networks, real-time implementation

## Abstract

Cogitive radio networks (CRNs) require high capacity and accuracy to detect the presence of licensed or primary users (PUs) in the sensed spectrum. In addition, they must correctly locate the spectral opportunities (holes) in order to be available to nonlicensed or secondary users (SUs). In this research, a centralized network of cognitive radios for monitoring a multiband spectrum in real time is proposed and implemented in a real wireless communication environment through generic communication devices such as software-defined radios (SDRs). Locally, each SU uses a monitoring technique based on sample entropy to determine spectrum occupancy. The determined features (power, bandwidth, and central frequency) of detected PUs are uploaded to a database. The uploaded data are then processed by a central entity. The objective of this work was to determine the number of PUs, their carrier frequency, bandwidth, and the spectral gaps in the sensed spectrum in a specific area through the construction of radioelectric environment maps (REMs). To this end, we compared the results of classical digital signal processing methods and neural networks performed by the central entity. Results show that both proposed cognitive networks (one working with a central entity using typical signal processing and one performing with neural networks) accurately locate PUs and give information to SUs to transmit, avoiding the hidden terminal problem. However, the best-performing cognitive radio network was the one working with neural networks to accurately detect PUs on both carrier frequency and bandwidth.

## 1. Introduction

Cognitive radio (CR) is a concept involving a communication device capable of knowing the spectral behavior in its environment and adapting to it. Taking advantage of spectral gaps (or holes) not utilized by licensed users, also known as primary users (PUs), CR technology allows nonlicensed or secondary users (SUs) to detect these available parts of the spectrum [1]. Specifically, the operation of CR involves four stages or functions: spectrum sensing, spectrum sharing, spectrum decision, and spectrum mobility. Spectrum sensing (SS) is a fundamental task of detecting one or more PUs; this stage shows whether the sensed spectrum is occupied or empty [2]. Usually, this task is carried out in single bands; however, the current paradigm of multiband spectrum sensing (MBSS) involves multiple bands that are not necessarily contiguous [3].

Many MBSS techniques issued from digital signal processing and machine learning (ML), such as wavelets, compressed sensing, energy detectors, and blind or semiblind methods, have been proposed, primarily in simulated scenarios [4,5,6,7,8,9,10]. Other works have been implemented in a real wireless communications environment [11,12,13,14,15,16], where software-defined radio (SDR) and universal software radio peripheral (USRP) technologies have been employed. Recently, SDR devices such as the HackRF One, the LimeSDR Mini, and the RTL-SDR have become extremely popular because of their affordability and good performances [17,18]. These generic communication devices offer radio equipment the flexibility of a programmable system, allowing the modifications to the communication system behavior simply by changing its software, permitting anyone, including hobbyists on a budget, access to the full radio spectrum.

In addition to knowing the PUs’ behavior in a given frequency domain and avoiding the hidden terminal issue, knowledge of the behavior of the radio spectrum in its specific geographical area of influence is necessary. For this reason, the idea of including a geographic tool constructed with the radioelectric information provided by the SUs issued from a real environment is pertinent. Under this perspective, the ability to build a radio environment map (REM) has become very important in recent years. A REM is “a tool that combines information collected from the radio environment, such as received signal intensity, interference measurements, propagation conditions, etc., for specific locations and frequencies, with the aim of building a map that provides an overview of coverage of the network” [19]. Thus, REMs permitting the characterization of the position, directivity, power, and modulation type of PUs have become a challenging task in cognitive radio network (CRN) design [20]. Indeed, in [21], REMs were used to locate relevant PUs in a geographic region of interest, characterizing their positions, directivities, powers, and modulation types. Likewise, in [22], REMs were used to sense the spectrum based on an adaptive compressed spectrum-sensing algorithm, contributing spatial information to the network capable of adapting to the radio environment. REMs are very flexible tools, as shown in [23], where they are combined with ML to determine the effective coverage area perceived by a cognitive sensor network, correctly estimating it at around 92%.

Another important tool, owing to the computing power and the amount of available data, is neural networks (NNs), used widely from pattern recognition and image classification to financial market behavior prediction and autonomous vehicle driving [24]. For spectrum sensing, for example, in [25], an NN was implemented to obtain the local information on single-node spectrum detection (spatial and temporal features). The information (extracted features) from multiple nodes fed another NN, thus permitting cooperation in the CRN. On the basis of REM and NN paradigms, this work proposed a new CRN methodology, implementing an MBSS involving a network integrated with low-cost SDR devices in a controlled, realistic wireless communications environment.

This work developed a CRN for monitoring multiband spectrum and locating PUs with their characteristic bandwidth, carrier frequency, and power in specific areas. In addition, our CRN located the spectral opportunities where the SUs could be located and, unlike models of other studies, could be implemented in real time. In the following, Section 2 briefly lays out the theoretical basis of REMs and NNs, and Section 3 presents the authors’ previous work in implementing the MBSS used as a basis for the new proposal presented in this work. Section 4 explains in detail the new proposed methodology, and Section 5 details the implemented real scenario. Finally, Section 6 offers the results, conclusions, and discussion [20].

## 2. Theoretical Basis of REMs and NNs

### 2.1. Construction of Radio Environment Maps

REMs are used in telecommunications research to represent the power distribution at a specific area emitted by different radio sources (one or more). A REM is constructed by collecting power measurements at different points in a specific area and interpolating them to produce a graphical representation (map) of the power distribution (or coverage) of the emitted signals, as shown in Figure 1. These maps are generally used to evaluate and predict the behavior of radio signals in a given environment, which is essential for capacity planning, communication system design, and troubleshooting related to interference. In addition, REMs are also used in CR applications, where devices can use the information provided by the REM to select the most suitable frequency to transmit data and avoid interference with other devices in the same location [20].

Collecting and interpolating power measurements are the two essential processes in constructing a *correct* map. In this case, interpolation methods [26] are critical for *correctly* estimating an unknown value between two or more known points (measurement points). In other words, a REM is simply a smoothed estimate of the power distribution based on a few known values of measured power. Two interpolation methods widely used to build REMs are inverse distance weighting (IDW) and Kriging, and both are explained below.

#### 2.1.1. IDW Method

The inverse distance weighting interpolation method [27,28] is simple and easy to implement, and it is widely used in applications such as precipitation estimation, topographic data interpolation, and air pollution estimation. IDW is a classical interpolation method in spatial analysis and is most commonly used in geostatistical and mathematical interpolation [29].

IDW estimates the value of a variable at a specific point based on known values at nearby points. This method rests on the idea that nearby points have a greater impact on the estimate than more distant points. Thus, IDW uses a formula that assigns a weight to each known point based on the distance between them and the unknown point. The weights are determined by considering the inverse of the distance, so that the closest points have higher weights and the farther ones have lower weights. For the IDW method, we considered the data transposed vector z=zs1,…,zsnT as observations from a random process. In our case, z contained the average power at each point of the considered sampled region (see Figure 1)
(1)zs:s∈D;D⊂ℝ2,
at known geographic locations s1,…,sn. The final estimate was obtained using the weighted sum of the known values in the nearby points, given by [28]:(2)Z(s)=∑sidsi,spwisi∑1dsi,sp,
where Zx is the estimated value of the variable at the unknown point s, zi is the known value of the variable at the *i*-th point si, dsi,s is the Euclidean distance between the unknown point s and the point si, p ∈ ℝ is a smoothing parameter controlling the influence of the known points over the estimated values, and wisi is the weight assigned to each known point si in order to account for the uncertainty in the known data. In practice, a value of p=2 or p=3 is usually used. The estimated values of Zs result in a mesh of unknown points. However, this method has limitations, such as a tendency to smooth out data variability and produce an inaccurate estimate in areas with few known points. In this work, Zs represents the power spectral density (PSD) in the different points around the sensed area.

#### 2.1.2. Kriging Method

The Kriging method [30] is an interpolation technique derived from regionalized variable theory. It depends on expressing the spatial variation of the property using the variogram, and it minimizes the estimated prediction errors [31]. The goal of Kriging is to find the most accurate and least uncertain estimate by considering not only the value of the variable at the known points but also the distribution of the variable in space and its correlation with the known points. Following [30], the Kriging method can be established by taking the points and the region mentioned in $\left(1\right)$, assuming μ is known and
(3)$$z\left(s\right)=\mu +\delta \left(s\right);s\in D,$$
are considered with known covariance function:(4)$$C\left(s,u\right)\equiv \mathrm{cov}\left(z\left(s\right),z\left(u\right)\right);s,u\in D,$$
where $\delta \left(\cdot \right)$ is a zero-mean stochastic process. Thus, the best linear unbiased predictor $z\left(s_{0}\right)$ is obtained by minimizing [32]
(5)$$E{\left(z\left(s_{0}\right)-\sum_{h=1}^{n}\lambda_{h}z\left(s_{h}\right)\right)}^{2},$$
with $\lambda_{1},\ldots ,\lambda_{n}$, subject to
(6)$$\sum_{h=1}^{n}\lambda_{h}=1.$$

The optimal values were computed using the method of Lagrange multipliers. For this, the mean-squared prediction error (Equation (5)) was obtained by
(7)$$\begin{array}{l} E{\left(z\left(s_{0}\right)-\hat{z}\left(s_{0}\right)\right)}^{2}=\\ =C\left(s_{0},s_{0}\right)-c^{\prime }C^{-1}c+{\left(1-c^{\prime }C^{-1}1\right)}^{2}{\left(1^{\prime }C^{-1}1\right)}^{-1}, \end{array}$$
where
(8)$$c\equiv {\left(C\left(s_{0},s_{1}\right),\ldots ,C\left(s_{0},s_{n}\right)\right)}^{\prime },$$
(9)$$C\equiv \left(C\left(s_{h},s_{j}\right)\right),$$
and
(10)$$\hat{z}\left(s_{0}\right)=\gamma^{\prime }\;\Gamma^{-1}Z+\left(1-\gamma^{\prime }\Gamma^{-1}1\right){\left(1^{\prime }\Gamma^{-1}1\right)}^{-1}\left(1^{\prime }\Gamma^{-1}Z\right),$$
(11)$$\begin{array}{l} E{\left(z\left(s_{0}\right)-\hat{z}\left(s_{0}\right)\right)}^{2}=\\ =\gamma^{\prime }\Gamma^{-1}\gamma -{\left(1-\gamma^{\prime }\Gamma^{-1}1\right)}^{2}{\left(1^{\prime }\Gamma^{-1}1\right)}^{-1}, \end{array}$$
where $\gamma \equiv {\left(\gamma \left(s_{0},s_{1}\right),\ldots ,\gamma \left(s_{0},s_{n}\right)\right)}^{\prime }$, Γ is an n×n matrix whose ${\left(i,j\right)}^{th}$ element is $\gamma \left(s_{i},s_{j}\right)$, and
(12)$$2\gamma \left(s_{i},s_{j}\right)\equiv C\left(s_{i},s_{i}\right)+C\left(s_{j},s_{j}\right)-2C\left(s_{i},s_{j}\right),$$
is the variogram [$\gamma \left(s_{i},s_{j}\right)$$\gamma \left(s_{i},s_{j}\right)$, named the semivariogram [33]]. The variogram, a valuable tool in modeling spatial variables, describes how the data are correlated with distance and, in doing so, allows us to accomplish spatial interpolation using the sampled data and the variogram information to estimate the variance of the values of the variable at the unsampled points [33,34]. In summary, Kriging was more precise than other interpolation methods, such as IDW or cubic interpolation [29], and suitable for applications where the data have a certain spatial dependence.

### 2.2. Neural Networks

Neural networks are artificial intelligence models inspired by the structure and functioning of the human brain. These networks comprise many nodes, known as „neurons”, connected to each other through „synapses”. Each neuron receives inputs from other neurons, processes them through an activation function, and sends its output to other neurons, as shown in Figure 2. The combination of inputs and connections between neurons allows a neural network to learn complex tasks from the selected training data. Neural networks are a powerful tool for machine learning because they can model complex relationships between inputs and outputs and make accurate inferences from real data. However, these tasks require a large amount of training data and are computationally intensive (sometimes impossible) to train and use in real-time implementations [35].

In this work, we used the multilayer perceptron (MLP), as shown in Figure 2. This type of artificial neural network consists of an input layer, one or more hidden layers, and one output layer. In the MLP, neurons in each layer are connected to neurons in the next layer, and they use an activation function to determine the neuron’s output. Information flows only in one direction, from the input to the output layers, passing through the hidden intermediate layers between the input and the output [36]. The general formula for calculating the output of a multilayer perceptron is given by [37]
(13)$$O=f\left(\sum \left(W_{i,j}\;I_{i}\right)+b\right),$$
where O is the output of the multilayer perceptron, f is the activation function, which can be a step function, sigmoid, rectified linear unit (ReLU), etc., $W_{i,j}$ are the synaptic weights connecting the input $I_{i}$ to the current neuron, $I_{i}$ is the input of the multilayer perceptron, and b is the bias, where i=1,…,n and j=1,…,k, with n being the number of input features to the NN (corresponding to the number of neurons in the input layer), k the number of neurons in a given hidden layer, and m the number of provided outputs (i.e., the number of neurons in the output layer of the NN).

The formula is applied to every neuron in every network layer, including the input, hidden, and output layers. Each neuron in a hidden layer takes the output of all the neurons in the previous layer (i.e., assuming a fully connected network) as its input, and the output of one neuron in the output layer is the network’s final output. During the training phase, the weights and the bias are adjusted to minimize the error between the network output and the desired output. This is achieved by a learning algorithm, such as the backpropagation algorithm [38], which updates the weights and bias in the direction of the downward gradient [39].

## 3. Preliminary Work

This section briefly describes the preliminary work on a novel MBSS technique based on the sample entropy (SampEn) developed and published by the authors in [39], constituting the basis of the new proposal. However, the MBSS technique presented in [39] was the result of the authors’ previous efforts to refine and complement it with modules implemented in real time to, for instance, diminish the noise introduced by SDR devices. Hence, several comparative studies have been carried out in order to measure the performance of the proposed MBSS technique compared with other classical methods, such as energy detectors. The efficiency of locating the PUs by this MBSS technique stands out in SNR values close to 0 dB [11]. Table 1 highlights the principal contributions of these previous works.

The last MBSS technique from Table 1 was implemented in a real environment using low-cost SDR devices. Each connected SDR device was considered an SU, independent from others, processing its information locally in a determined single band to achieve ensemble wideband spectrum sensing. The MBSS technique used in this proposal, however, combines different SDR devices, namely the RTL-SDR [42], the HackRF One [43], and the LimeSDR mini [44], to conform each considered SU, forming a CR infrastructure, which permits having an MBSS in a specific location. Figure 3 below shows how this MBSS method works. The MBSS blocks are as follows:

**Sliding window of 100 ms.** In this block, the complex signal $x_{Ii,l}(n)+jx_{Qi,l}(n)$ from each SDR in the time domain was received and updated every 100 ms from the radio environment of the *i*-th SU integrated by *z* different SDR devices;**Power Spectrum Density (PSD) estimation.** In this module, the Welch method [45] was applied to each signal $x_{Ii,l}(n)+jx_{Qi,l}(n)$ in order to obtain, on a linear scale, the wideband PSD $R_{i,l}(k)$ from the SU ensemble;**Impulsive noise reduction.** In this block, impulsive noise, high-frequency noise, and abrupt changes (many of them generated by the SDR devices themselves) in the signal $R_{i,l}(k)$ were eliminated or diminished through discrete wavelets via the multiresolution analysis [46], resulting in the signal ${R^{\prime }}_{i,l}(k)$;**Estimation of frequency bands and detection of primary users.** In this module, the MBSS technique based on SampEn, K-means algorithm [47] (permitting to optimize specific detection parameters), and discrete wavelets was applied to each SU’s analyzed wideband spectrum in order to obtain the spectrum occupation given by $Occupation_{T_{t}}\left\{{R^{\prime }}_{i,l}\left(k\right)\right\}$. This result included three vectors ${\left[b_{i,1},b_{i,2},\ldots ,b_{i,N-1}\right]}_{T_{t}}$ which contained binary values indicating occupied (“1”) and empty (“0”) bands. The second vector was ${\left[L_{i,1},L_{i,2},\ldots ,L_{i,N}\right]}_{T_{t}}$ and contained the corresponding computed boundaries for each detected band, and the third was ${\left[P_{i,1},P_{i,2},\ldots ,P_{i,N-1}\right]}_{T_{t}}$, which contained the power for each detected band.

One of the primary motivations for this technique was to propose an MBSS method (i) adequate to correctly detect a primary user and the operating parameters, such as carrier frequency, bandwidth, and power, with (ii) computational complexity permitting a real-time implementation. Both aims were fulfilled, and the results were used to create a compact CRN capable of sharing spectral information with a central entity in order to estimate the area occupied by different detected PUs in the studied zone. A wireless communications environment was proposed considering these SDR devices.

## 4. New Proposal and Methodology

This section presents the new research proposal and the implemented methodology.

### 4.1. Proposal

Our previous work’s novel idea was to consider a computer integrating different SDR devices detecting each PU in a single band to sense a wide frequency range, as shown in Figure 4a. This model was seen as the only entity containing different SUs. Our follow-up work proposed replicating this entity containing different connected SDR devices and creating a cooperative CR network for sensing a wide frequency range in a broader geographical region. Each entity was considered an SU containing different technologies for sensing a broad spectrum, as shown in Figure 4b. In addition, to sense the radio spectrum and uncover the PUs’ behavior in a given geographical region, this cooperative CR network had the task of avoiding the hidden terminal problem. One of the main strengths of this new proposal (see Figure 4b) over previous work is the cooperation of the different SUs in geographically locating the PUs through the REMs and the ability to carry out this processing in real time.

### 4.2. Methodology

The proposed methodology can be outlined in three big blocks, shown in Figure 5:The collection of information obtained by SUs. For this, each SU locally processes the sensed data to be sent to a central entity, including the occupancy of the observed spectrum in its geographic location and the frequency band edges and estimated power vectors;The database for storing the information obtained by each SU at a specific time;The central entity overseeing the processing determines the geographic area occupied by the detected PUs in the radio spectrum.

Each block is described in detail below.

#### 4.2.1. Collection of Information Locally by the SUs

The information collection was carried out every 100 ms for each SU, resulting in the $Occupation_{T_{t}}\left\{{R^{\prime }}_{i,l}\left(k\right)\right\}$ of analyzed spectral signal, as mentioned in Section 3 and displayed in Figure 3. This information was conveyed by three vectors: (i) the edge detector vector ${\left[L_{i,1},L_{i,2},\ldots ,L_{i,N}\right]}_{T_{t}}$, which stores the frequency edges where it is possible to find the presence of PUs, (ii) the binary decision vector ${\left[b_{i,1},b_{i,2},\ldots ,b_{i,N-1}\right]}_{T_{t}}$, which stores a binary decision for corresponding delimited bands where noise or a possible PU transmission is detected, and (iii) the power vector ${\left[P_{i,1},P_{i,2},\ldots ,P_{i,N-1}\right]}_{T_{t}}$, which stores the received average power corresponding to each classified window (i.e., each binary decision). The vectors in (ii) and (iii) are the same size.

#### 4.2.2. Database

The edge detector, binary decision, and power vectors were stored in the database. It is important to note that the information stored in the database stemming from each SU was not necessarily similar; the lengths of the vectors were different for each SU. Indeed, each SU observed a different behavior of the radioelectric spectrum, because they were placed randomly in different geographic coordinates. To account for this discrepancy, all vectors were labeled with the exact sensing time *T_x_* (see Table 2), i.e., synchronized. Information was uploaded to a server, saving only enough relevant data (three different vectors for each SU) in the database to permit fast storage and extraction, taking advantage of SUs’ hardware performing the MBSS technique locally. Indeed, the three uploaded vectors for each SU take up very little memory space. In the best case (without even a PU in the sensed spectrum), the edge detection vector had a length of two, while the binary decision vector and the power vector had a length of one. In the worst case, when the SNR had a low value (close to 0 dB), multiple windowing appeared due to noise, thus increasing the vectors’ lengths. However, regardless of the size of the three vectors, the information was compact and facilitated the reconstruction of the spectrum signal occupation, including the average power for each reconstructed window.

#### 4.2.3. Central Entity

The main tasks of the central entity were (i) to indicate how many PUs appear in the spectrum compared with the perceived SUs, (ii) to build a REM for each detected PU using the location of each SU, the perceived power in each SU, the time of radio space monitoring as parameters, and finally (iii) to show the area covered by the PUs.

Two ways of implementing these tasks were developed, the first through *classical* digital processing and the second using machine learning techniques (specifically, the NN). Both approaches are detailed below.

##### 4.2.3.1. Central Entity Implementing Classical Digital Signal Processing

Digital signal processing, which uses programmed algorithms to process and analyze data, is a well-established and stable technology that has been used for decades in a variety of applications. It is relatively easy to implement and can be very fast and efficient in situations where the datasets and tasks are specific. In this case, the central entity was implemented with classical techniques for reconstructing the spectrum occupation from the shared information (edge detector, binary decision, and power vectors) obtained by each SU in the database. The central entity processing the information is shown in Figure 6 and described by Algorithm 1.

**Algorithm 1** Central Entity Computed by Traditional Signal Processing

**Step 1.1**
The central entity collects the edge detector, binary decision, and power vectorsof each PU at determined time instant t from the database. With this information, the central entity reconstructs the occupancy signal OccupationTtR′ik and constructs simultaneously, an approximation of the power spectral density PSD_recTtR′ik using only the average powers (power vector) associated to each SU. The length of these frames is set to 1024 samples.
**Step 1.2**
Once the reconstruction of the occupation of each SU is performed, the mean $\mathrm{value}\;E\left[Occupation_{T_{t}}\left\{{R^{\prime }}_{i}\left(k\right)\right\}\right]$ is computed.
**Step 1.3**
After computing the average value of the Occupation, it is possible to infer the presence of one or several PUs, giving two possible vectors. First, the vector $F_{C\_vector}=\left[F_{C1},F_{C2},\ldots \right]$ containing the central frequencies values of detected PUs. The length of this vector will be the number of PUs detected by the algorithm. A second vector, containing the occupied bandwidths by each $\mathrm{detected}\;\mathrm{PU}\;B_{vector}=\left[B_{PU_{1}},B_{PU_{2}},\ldots \right]$, is also obtained. Through the number of singularities and their corresponding widths detected in $E\left[Occupation_{T_{t}}\left\{R_{i}{}^{\prime }\left(k\right)\right\}\right]$, it is possible to estimate how many PUs are in the $\mathrm{spectrum}\;\mathrm{and}\;\mathrm{their}\;\mathrm{respective}\;\mathrm{bandwidths}\;(B_{vector}=\left[B_{PU_{1}},B_{PU_{2}},\ldots \right]$). The $F_{C\_vector}$ is integrated by the central frequency values of estimated bandwidth for each detected PU. 
**Step 1.4**
Knowing now the central frequency and the corresponding bandwidth of each detected PU, it is possible to locate them in each reconstructed PSD $PSD\_rec_{T_{t}}\left\{R_{i}{}^{\prime }\left(k\right)\right\}$$.\;\mathrm{The}\;\mathrm{mean}\;\mathrm{of}\;\mathrm{the}\;\mathrm{interval}\;B_{PU_{N}}$$\;\mathrm{centered}\;\mathrm{on}\;F_{C_{N}}$ of the $PSD\_rec_{T_{t}}\left\{R_{i}{}^{\prime }\left(k\right)\right\}$$\;\mathrm{is}\;\mathrm{computed}\;\mathrm{giving}\;\mathrm{the}\;\mathrm{scalar}\;aux\_psd_{i,N}$. In the strict sense, this scalar represents the average power in the carrier, sensed by each $\mathrm{SU},\;\mathrm{where}\;\mathrm{the}\;PU_{N}$ is or should be.
**Step 1.5**
With the scalar aux_psdi,N of each SU located in a specific coordinate, the REMwill be built using a double interpolation. First interpolation is done through the IDW method and for the second one, the Kriging method is applied. Due tothe fact that in our case only eight geographical points are considered (i.e., onlyeight SUs are implemented), this double interpolation is carried out with the purpose of having a better precision to describe the behavior of the radio electric space in the environment described later. In this case, each REM is constructed with values aux_psdi,N that specifically correspond to the average power of the bandwidth BPUN.
**Step 1.6**
Finally, the active area of each PU will be determined according to the information collected by each secondary entity SUi. For this, a threshold of −80 dBm is used to classify the area estimated by the REMs that was chosen in[41] for a wireless environment. That is, regions in the REM that are above this threshold correspond to the active area of detected PUs.

##### 4.2.3.2. Central Entity Implementing Neural Networks

The central entity was implemented using ML techniques, specifically NNs. NNs have the ability to learn from data and improve their performance as they absorb more information. This makes them particularly useful in applications where the data are complex, difficult to understand, and processed using traditional methods. Furthermore, NNs can perform tasks that traditional methods cannot, such as recognizing patterns in unstructured data or processing input signals that change over time. In this case, some modules in the central entity were replaced by NNs, as shown in Figure 7. The steps of this operation (Algorithm 2) are described below.

**Algorithm 2** Central Entity Computed by NNs

**Step 2.1**
The central entity collects the data from each of the SUs in the database at a $\mathrm{specific}\;\mathrm{time}\;T_{x}$. In the module input estimator and processor, a processing is carried out in order to estimate the inputs driving a NNs ensemble. As $\mathrm{mentioned}\;\mathrm{above},\;\mathrm{the}\;\mathrm{three}\;\mathrm{vectors}\;\mathrm{coming}\;\mathrm{from}\;\mathrm{each}\;\mathrm{SU}\;\mathrm{for}\;\mathrm{time}\;T_{x}$ do not $\mathrm{have}\;\mathrm{the}\;\mathrm{same}\;\mathrm{length}\;\mathrm{as}\;\mathrm{the}\;\mathrm{vectors}\;\mathrm{for}\;\mathrm{time}\;T_{x+t_{1}}$$\;\mathrm{or}\;T_{x-t_{2}}$. Even, the size of the vectors differs from one SU to another SU even though they are at the same $\mathrm{time}\;T_{x}$. Based on this fact, this module oversees building the input vectors $PSD\_rec_{T_{t}}\left\{R_{i}{}^{\prime }\left(k\right)\right\}$$\;\mathrm{and}\;Edge\_rec_{T_{t}}\left\{R_{i}{}^{\prime }\left(k\right)\right\}$ driving the NNs. For this, these vectors must have always the same length (in our case this length is set to 13 samples). This block is detailed below.
**Step 2.2**
The vector PSD_recTtRi′k is evaluated by NN1, at the same time the vector Edge_recTtRi′k is evaluated by NN2 and by NN3. This evaluation is performed sequentially, i.e., each vector of each SU will be evaluated by its corresponding NN one after the other until the result of the i-th connected SU be obtained. As a result of this step, NN1 provides an approximate number of detected PUs and their powers. NN2 gives an approximate number of detected PUs and their central frequencies. Finally, NN3 returns an approximate number of detected PUs and their bandwidths. At the end of this section, it is discussed what would happen if the number of PUs detected by each NN is not the same.
**Step 2.3**
The information obtained from Step 2.2 is evaluated to determine the number of PUs in the spectrum and their corresponding power, bandwidth, and carrier frequency. This evaluation is detailed below.
**Step 2.4**
The information obtained from Step 2.3 will be shared with the REM estimator module. This block receives (i) the geographic coordinates of the SUs in the network, (ii) the data of the possible PUs in the spectrum (power, carrier, and $\mathrm{bandwidth})\;\mathrm{and}\;(\mathrm{iii})\;\mathrm{an}\;\mathrm{identifier}\;\mathrm{that}\;\mathrm{corresponds}\;\mathrm{to}\;\mathrm{time}\;T_{x}$ in which the spectrum was monitored.

The rest of the steps of this algorithm that include NNs corresponds to Steps 1.5 and 1.6 of the methodology described in Algorithm 1 (Section 4.2.3.1). The *input estimator and processor* (Step 2.1) and the *information collector* (Step 2.3) modules are described below.

Figure 8 shows the process of generating $PSD\_rec_{T_{t}}\left\{{R^{\prime }}_{i}\left(k\right)\right\}$, the input vector of $NN_{1}$. First, power, edge detector, and binary decision vectors were used, with samples ranging in the intervals of [1–20], [1–20], [2–21], respectively, according to the studies carried out in our preliminary work [39]. As mentioned above, using these vectors, we obtained (i) the frequency range that was sensed by each SU, (ii) the number of samples of the estimated power spectral density, and (iii) the number of windows in which the sensed spectrum was partitioned. An occupation value was assigned for each detected window, 1 for a possible PU transmission or 0 for the noise. Accompanying each occupation value was an average power value or the power vector element representing the power estimate in each detected window. The occupation and the power signal were reconstructed using these average values of each detected window, setting both vectors to a length of 1024 samples each.

After that, the power- and occupation-reconstructed vectors were multiplied column by column, resulting in the Power_Occupation vector of 1024 samples. In this last vector, locations with estimated noise had a value of 0, and those with the possible transmission of one or several PUs had a value corresponding to the average power of detected windows. A downsampling of the Power_Occupation vector was performed to reduce it to only 10 samples and to combine it with the coordinates of the SUs (to which the analyzed vectors correspond) and the time $T_{x}$ at which the signal $Occupation_{T_{t}}\left\{{R^{\prime }}_{i,l}\left(k\right)\right\}$ was sensed, thus finally integrating the input vector for $NN_{1}$. It should be noted that the time parameter is extremely important since the spectrum behaves differently over time.

Figure 9 shows the construction of the vector $Edge\_rec_{T_{t}}\left\{{R^{\prime }}_{i}\left(k\right)\right\}$. Here, the occupancy and the frequency bands vectors corresponding to the frequency interval perceived by the corresponding SU were reconstructed. This vector was also multiplied column by column with the occupation vector, resulting in the Freq_Occupation vector being 1024 samples in length. This latter vector was assigned zeros where it corresponded to noise and nonzero values for the samples that corresponded to one or several PU transmissions. Again, downsampling was applied to reduce this vector to only 10 samples and join it with the coordinates of the SUs and the time $T_{x}$ corresponding to the sensing period. This concatenation drives the input of $NN_{2}$ and $NN_{3}$.

The $NN_{1}$ output corresponds to the power of each PU detected by each SU. When the PU power value was less than −80 dBm, it was interpreted as no transmission, corresponding to noise. The output of $NN_{2}$ resulted in the carrier on which a possible PU was located. Finally, the $NN_{3}$ output delivered the transmission bandwidths corresponding to each detected PU in the analyzed spectrum.

The resulting outputs from NNs for each SU are shown in Figure 10. These outputs can be classed into three large blocks: powers, carrier frequencies, and transmission bandwidths of each PU detected by the corresponding SU. The *information collector* block analyzes the output of each SU making up the network and sends it to the module permitting the REM estimate. For example, Figure 10a shows that the power of $PU_{1}$ is less than −80 dBm; the *information collector* interprets that $PU_{1}$ does not correspond to a transmission and, by sharing the information delivered by $SU_{1}$ with the estimator of REM, assumes that $SU_{1}$ observes (j−1) PUs (where *j* represents the number of possible PU transmissions). Figure 10b shows that the carrier frequencies of $PU_{1}$ and $PU_{j}$ are outside the monitored space, so the *information collector* shares with the REM estimator that $SU_{2}$ only observes (j−2) PUs. In Figure 10c, the B of $PU_{1}$ is a very small value (possibly corresponding to impulse noise); thus, the *information collector* shares with the REM estimator module that the $SU_{n}$ observes (j−1) PUs. It is important to point out that any of these combinations shown in Figure 10 may change the number of observed PUs, i.e., if the power of a first PU does not exceed the threshold of −80 dBm, the carrier of a second PU is not in a correct frequency range, and the B of a third PU tends to zero; then, these three PUs will be considered as noise.

The proposed methodology has been implemented in a real wireless communication environment. This controlled environment is explained in the next section.

## 5. Real Wireless Communication Environment

Figure 11 shows the real, controlled environment implemented in our research. In this proposed scenario, we considered two PUs collocated at the center of the studied area while, at the same time, eight SUs were sensing their behavior in their geographical zone of influence. It is important to note that the SUs and PUs did not share the geographic coordinates among themselves, nor was that information used in spectrum sensing. PUs were located at the center, hoping most SUs could receive part of the Pus’ signal. SUs were set randomly in the area of study.

Table 3 specifies the involved parameters of both SUs and PUs. These SUs shared the channel occupancy with the database and the central entity in order to determine (i) how many PUs on average were observed in this environment, (ii) the B and the $F_{C}$ in which they located the detected PUs, and (iii) finally the area occupied by the detected PUs.

PUs and SUs were deployed in an area 12×12 m^2^, containing structures such as walls, doors, windows, columns, etc., as indicated in Figure 11.

## 6. Results

This section outlines the results of implementing the proposed methodology in a real wireless communication environment. They are divided into two subsections; the first corresponds to the central entity based on digital signal processing, and the second concerns the central entity based on NNs.

### 6.1. Results with a Central Entity Based on Digital Signal Processing

This section shows the results of the implemented scenario presented in Figure 11. Figure 12 shows the reconstruction of the occupation of each SU by the central entity. In this case, we can highlight the following important points:

There were SUs who failed to perceive both PUs;The central frequency $F_{C}$ of each PU was perceived at a different frequency by each SU;The bandwidth size B for each PU varied for each SU;SUs that were the farthest from the PUs failed to detect both PUs.

Figure 13 shows the result of constructing an approximation of the PSD for each SU, considering only mean power values for each spectrum section. These approximations were used for the estimation of the REM.

Figure 14 shows the result of applying the module for estimating the average occupancy of each SU in addition to the result of the $F_{C}$ and B estimation for each PU. The first value obtained was $F_{C1}=699.48$ MHz with a bandwidth of $B_{1}=0.4$ MHz. $F_{C2}=700.49$ MHz with a bandwidth of $B_{2}=0.825$ MHz was also obtained. Given that the exact values were $F_{C1}=699.5$ MHz, $B_{1}=0.5$ MHz, $F_{C2}=700.5$ MHz, and $B_{2}=1$ MHz, the values we obtained here show a good occupancy estimation of the MBSS method in conjunction with the central entity for a specific geographical zone covered by the PUs.

Figure 15a and Figure 16a show the REM of $PU_{1}$ and $PU_{2}$, respectively. These maps were created based on the information collected by the SUs in their different locations. Figure 15b and Figure 16b show the respective areas occupied by $PU_{1}$ and $PU_{2}$. Values of the occupied area for each PU are $area_{PU_{1}}=60.76\;m^{2}$ and $area_{PU_{2}}=56.39\;m^{2}$. These results were obtained by placing a threshold L=−80 dBm in the obtained REM. In this way, the area with a power greater than this threshold was considered a space occupied by the PU.

Figure 15b shows that $SU_{6}$ and $SU_{8}$ did not observe the transmission of $PU_{1}$ (i.e., both $SU_{6}$ and $SU_{8}$ did not appear in this area of influence). This effect was a result of the structural distribution in which the implementation of the real scenario occurred. Nevertheless, an expected collaboration between SUs might improve this result.

### 6.2. Results with a Central Entity Based on NNs

This section details the results obtained using a central entity and NNs. First, the NN training is presented, and then the results obtained by applying ML algorithms to this stage are shown.

#### 6.2.1. Training

For the $NN_{1}$ training, $NN_{2}$ and $NN_{3}$, the backpropagation algorithm, the Levenberg–Marquardt activation function, 1000 epochs, the mean square error as loss function, and a low learning rate were used. In addition, 9000 vectors of inputs and their corresponding outputs were used. To determine the best NN?, the number of layers that should be used, and how many neurons, 12 NNs were studied to carry out the work of $NN_{1}$, $NN_{2}$, and $NN_{3}$. Each NN was considered with the number of layers la=1,2,3,4 and with the number of neurons ne=16,32,64. For each NN, there were 12 possible combinations between neurons and layers. They were performed to find the most convenient NN configuration for this work.

The results are presented for both the entity that uses digital signal processing and the one that uses the three NNs. As the first parameter, the training time used by each NN is shown in Table 4, indicating the time each NN spent in the training stage. Additionally, Figure 17 shows that the more neurons and layers an NN had, the longer its training time was. All the processing was carried out using the same computer (MacBook Pro with 8 GB RAM and a 1st-generation M1 processor), and the training time for each NN in its different versions tended to follow similar behavior.

#### 6.2.2. Statistics

This section analyzes the statistical results of the three NNs used in this proposal. To obtain these results, 27,000 entries were considered for each NN. Figure 18a presents the results of $NN_{1}$, responsible for granting the power value of the detected PUs, showing the power difference between the real values and those obtained with the different versions of $NN_{1}$. The difference between these values was, on average, −0.01 dBm for each version of $NN_{1}$, highlighting the accuracy of the predictive ability of the NN-based approach. Figure 18b only shows the average value, indicating that the margin of error between the expected value and those given by the different NNs was very small; the values were practically the same.

Figure 19 shows the result of $NN_{2}$, the network that gives the carrier value of each PU detected. In the proposed environment, the $NN_{2}$ detected two PUs, and Figure 19a shows its precision in detecting each PU carrier, with $F_{C1}$, on average, at 699.4 MHz, regardless of the configuration of $NN_{2}$. Figure 19b shows the precision of $NN_{2}$ in detecting the $PU_{2}$ carrier, indicating that, on average, the carrier value was 700.494 MHz. Even though the network with 2 layers and 64 neurons per layer deviated slightly from the ideal value (700.5 MHz) compared with the other NN configurations, it continued to provide reasonably accurate results. In both images, the value of applying digital signal processing can be seen in blue and is quite close to the ideal values of 699.4934 MHz and 700.4935 MHz for $PU_{1}$ and $PU_{2}$, respectively, thus resulting in a comparable level of accuracy of the NN-based approach.

Figure 20 presents the results of $NN_{3}$, the network in charge of giving the B that each detected PU in the spectrum frames. In this figure, the average B of $PU_{1}$ is 0.48 MHz, regardless of the network configuration (see Figure 20a). Moreover, Figure 20b shows that notwithstanding the NN structure, the value of the B for $PU_{2}$ is practically 1 MHz, which perfectly coincides with the ideal value (see Table 3). The blue dot in the image indicates the result of digital signal processing, as $B_{1}=0.425$ MHz and $B_{2}=0.824$ MHz. For $B_{1}$, the margins of error/precision of DSP and NN were quite similar (only DSP overestimates while NN underestimates). For $B_{2}$, however, it seems that the NN was much more precise, suggesting that the method based on NNs tends to improve DSP or, in the worst case, provide a comparable absolute error.

To measure the precision of this methodology, the F1 score (F1) [48] was used as a common evaluation metric in the ML field for assessing the accuracy of binary classification models. This metric combines the accuracy and recall of the model into a single measure.

Precision refers to the ratio of true positives (TP) to the sum of true and false positives (FP), while recall, on the other hand, refers to the ratio of true positives to the sum of true positives and false negatives (FN). The F1 value expresses the harmonic mean of precision and completeness, giving more weight to low values. The F1 formula is
(14)$$F1=2\frac{\left(precision*recall\right)}{\left(precision+recall\right)}$$

A value of F1 equal to 1 indicates that the accuracy and completeness are perfect, while an F1 of 0 indicates that the model is unable to correctly classify any of the samples. The F1 metric is valuable for comparing different binary classification models and selecting the best model for a given classification task. To determine this parameter (i.e., F1), the following four possible cases were considered (see Figure 21):

A window corresponding to a PU transmission and classified as such by the SU is considered a true positive (TP) value;A frequency window corresponding to a PU transmission classified as noise by the SU is a false negative (FN) value;A window corresponding to noise classified by the SU as a PU transmission is a false positive (FP) value;A frequency window corresponding to noise classified by the SU as such is a true negative (TN) value.

Figure 22 shows the F1 of the $NN_{1}$, which indicates whether the PUs were correctly located. In this image, all the NNs have a nearly perfect performance, around 0.98.

In the case of B, the F1 was estimated from the following cases (see Figure 23):

The resulting $NN_{3}$ B matching the ideal bandwidth that corresponds to a transmission is a true positive (TP) case;The resulting $NN_{3}$ B corresponding to an ideal bandwidth close to zero is a true negative (TN) case;The resulting $NN_{3}$ B much greater than an ideal bandwidth that is close to zero is a false positive (FP) case;The resulting $NN_{3}$ B that should correspond to an ideal transmission B but is a value close to zero will be a false negative (FN) case.

Figure 23 shows a ΔB value, which is the difference between real and estimated bandwidths when there is a PU. When this parameter tends to grow, it provides flexibility so that the resulting $NN_{3}$ B matches an ideal B that corresponds to one transmission. However, when the value of ΔB is very small, the system becomes more inaccurate, since it only detects as TP those values that are close to the ideal value.

Figure 24 shows the F1 of $NN_{3}$. The NN with four layers and the NN with two layers both had an undesirable performance. This result could be explained by observing Figure 20, which shows the result of $B_{1}$ is very close to the minimum value (ideal_value−ΔB). Nevertheless, this undesirable performance was observed only when a low number of neurons per layer (i.e., 16) were employed. By properly configuring the number of neurons per layer to a sufficiently high value (e.g., 32 or greater), accurate performance with F1 values around 0.95 and 0.96 was obtained, as well as with two and four layers in the NN. Figure 24 also suggests overfitting in the NNs with two and four layers. This is because the models for the 2-layer and 4-layer NNs, both with 16 neurons per layer, fit too well to the training data and, as a result, do not generalize well to the new data. Despite this, they have an F1 of 0.9 and 0.79, respectively.

Figure 25 shows the ratio of F1 scores between training time for $NN_{1}$ and $NN_{3}$ (Figure 25a and Figure 25b, respectively). The value obtained from this relationship indicates which NN provides the best trade-off based on the F1 accuracy result and the time invested in training the network to attain such accuracy. As can be noticed, using an NN with a single layer provides the best F1 performance for the training time required to obtain it. Further increasing the number of layers will increase the F1 score performance but will proportionately require a much longer training time, thus leading to a worse trade-off or relation between benefit (represented by F1 performance) and cost (represented by the required training time).

Finally, Figure 26 shows the behavior of the proposed methodology over time, during which the sensed spectrum was analyzed in a specific geographic area and where SUs collaborated to obtain the REM of the different detected PUs. As can be appreciated, the proposed methodology was able to characterize the dynamic temporal evolution of the REM in the geographical area of interest, thus providing a valuable tool for the study of CRNs.

## 7. Conclusions

In this work, a real wireless communication scenario was implemented to detect the occupancy of multiple PUs through several SUs via a central entity, permitting the determination of the area used by the PUs based on REM estimations. Apart from the REM constructions, other features, such as power, bandwidth, and central frequency of possible detected PUs from the multiband spectrum frames, were estimated by the considered SUs.

For this task, we proposed using neural networks to substitute the classical digital signal processing used in our preliminary work. It was expected that the performance and processing time would be faster. Clearly, the training stage of the NNs, as shown by the results, was a factor to be considered. In this work, higher precision was preferred to locate the PUs and less processing time for the central entity when using the NNs. However, it must be considered that NN training time was not negligible and could even be high, depending on the number of layers and neurons each NN contained. The difference between the real values and that of the NNs was minimal, and it could even be said they were practically the same. In some cases, the NN showed even better results than the digital signal processing, for instance, when detecting the PUs’ carriers.

Neural networks are a powerful and useful tool in many applications but are not always the best option. In some cases, classical digital signal processing may be sufficient, while in others, neural networks can significantly improve performance and accuracy. Future work will aim to determine the optimal number of SUs needed to obtain the area occupied by the PUs, thus avoiding the excessive processing of the central entity.

## Figures and Tables

**Figure 1 sensors-23-05209-f001:**
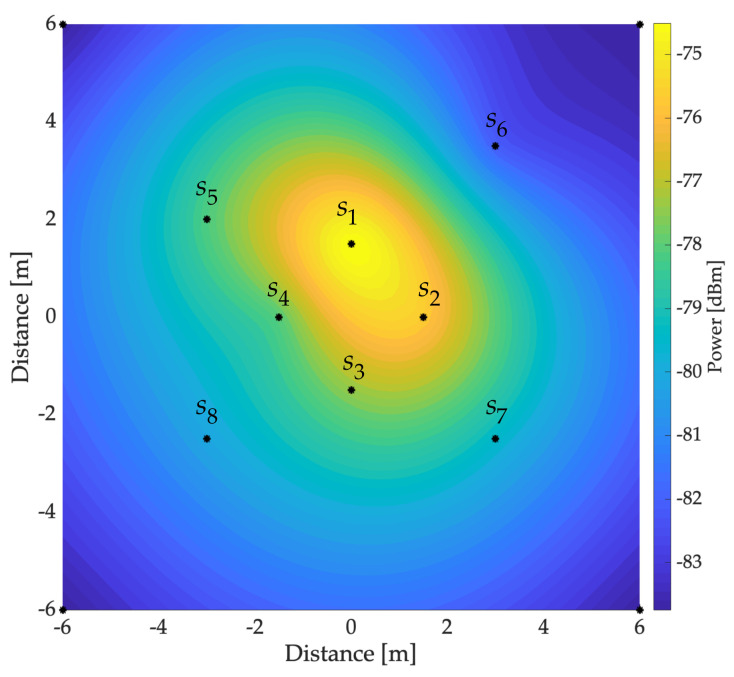
Example of a REM: Black dots indicate locations where the power measurements are taken using eight sensors. In this case, only a transmission source with −60 dBm is considered.

**Figure 2 sensors-23-05209-f002:**
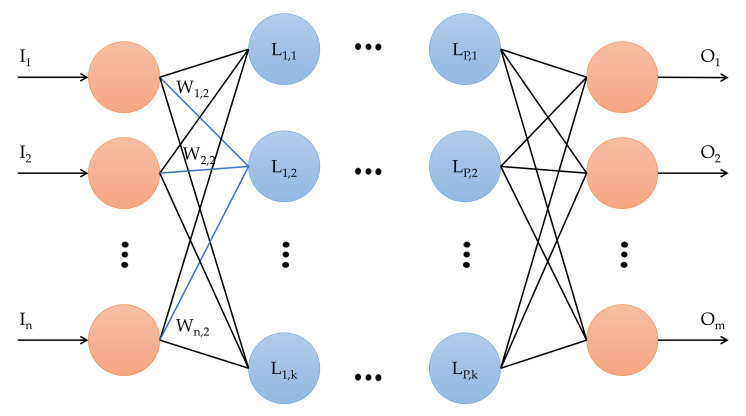
Neural network concept: the multilayer perceptron.

**Figure 3 sensors-23-05209-f003:**
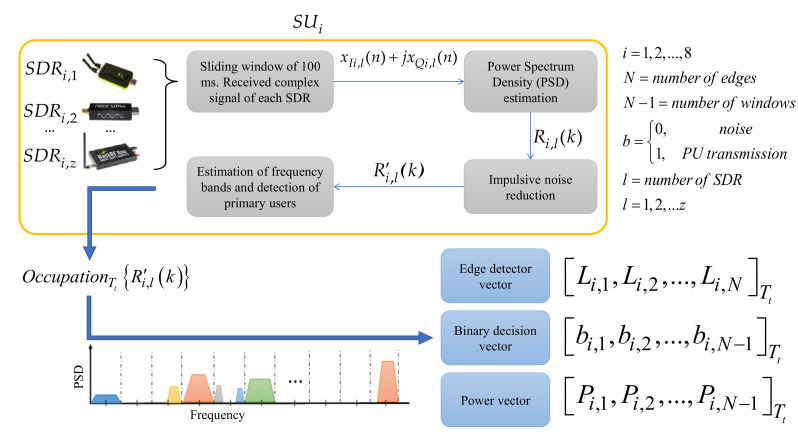
General scheme of implemented MBSS technique [40].

**Figure 4 sensors-23-05209-f004:**
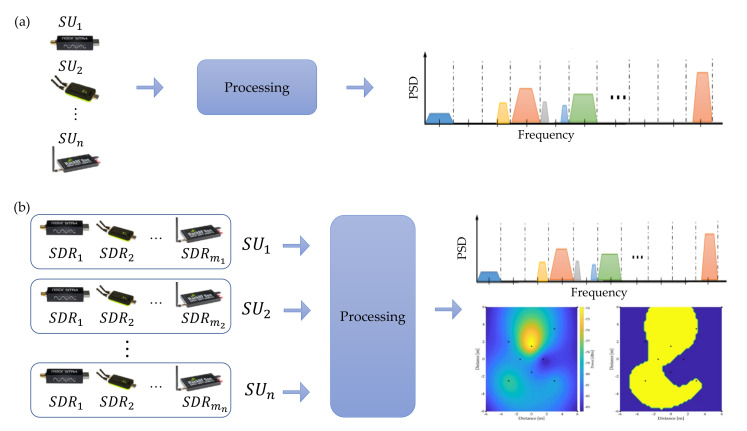
(**a**) Previous work and (**b**) new proposal.

**Figure 5 sensors-23-05209-f005:**
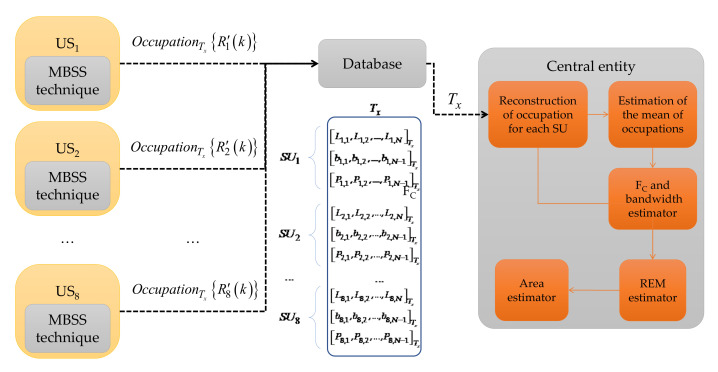
General scheme of the methodology.

**Figure 6 sensors-23-05209-f006:**
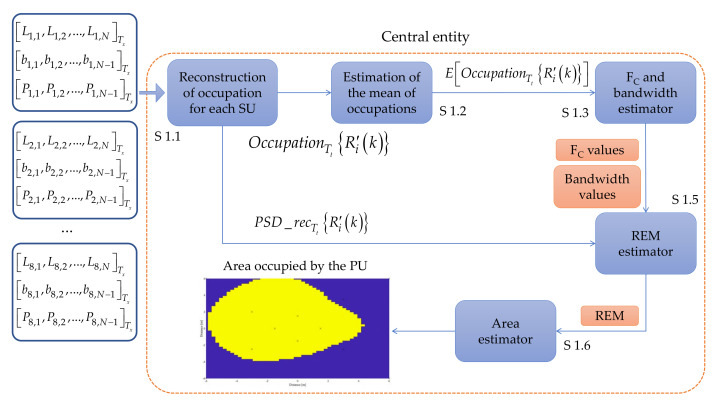
Implementation of the central entity using classical digital signal processing.

**Figure 7 sensors-23-05209-f007:**
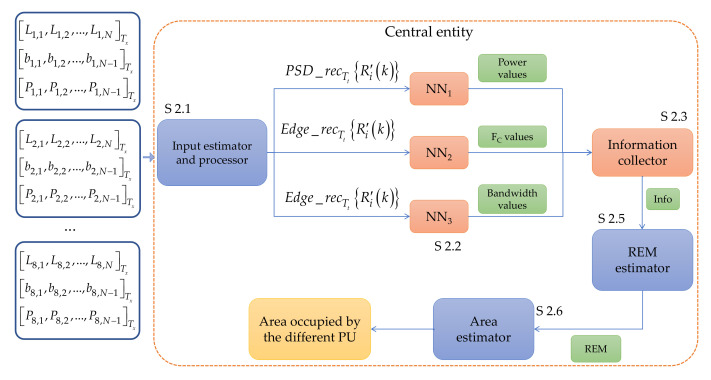
The central entity determining the REM construction parameters and area maps through NNs.

**Figure 8 sensors-23-05209-f008:**
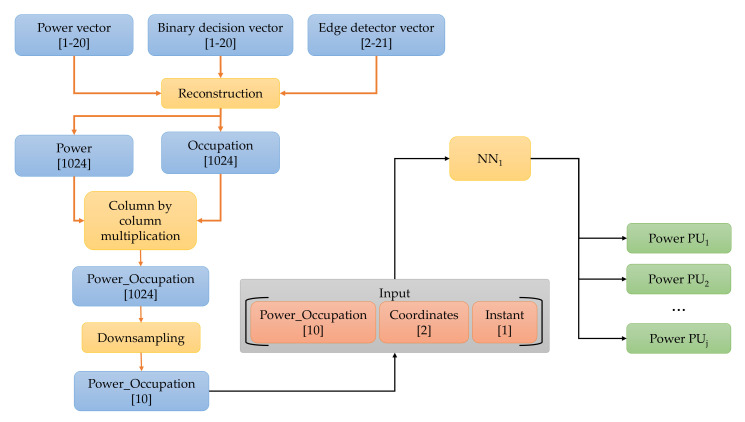
$Edge\_rec_{T_{t}}\left\{{R^{\prime }}_{i}\left(k\right)\right\}$ vector construction.

**Figure 9 sensors-23-05209-f009:**
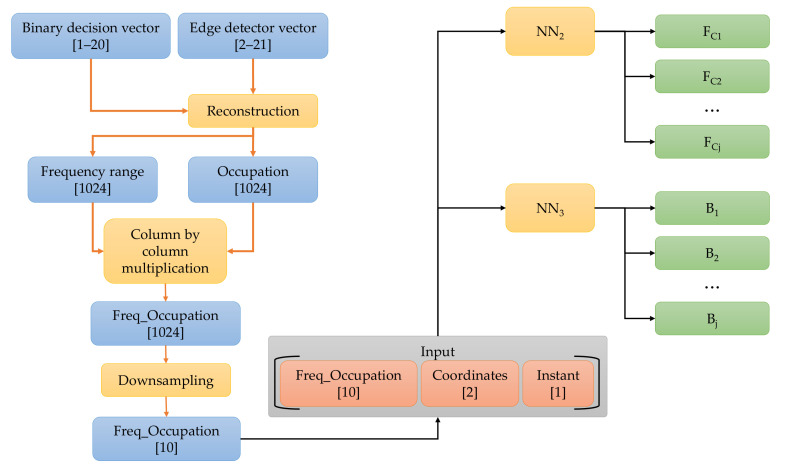
$Edge\_rec_{T_{t}}\left\{{R^{\prime }}_{i}\left(k\right)\right\}$ vector construction.

**Figure 10 sensors-23-05209-f010:**
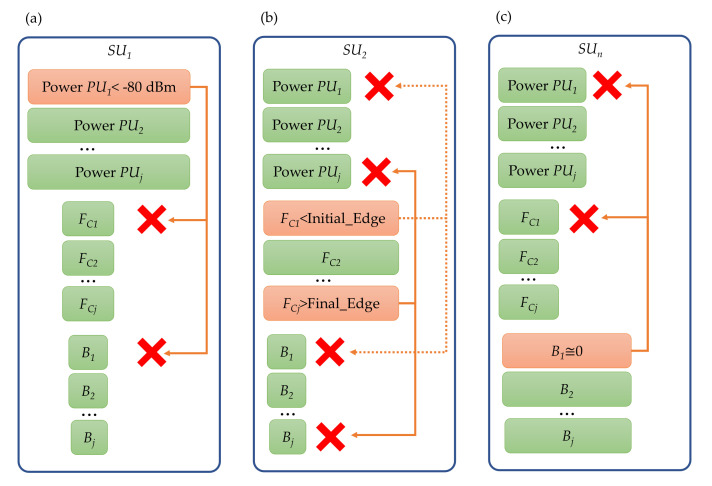
(**a**). The $SU_{1}$ observes j PUs, but $PU_{1}$ is discarded because it does not meet the threshold. (**b**) The $SU_{2}$ observes j PUs; nevertheless, $PU_{1}$ and $PU_{j}$ are discarded because they are not in the correct frequency range. (**c**) The $SU_{n}$ observes j PUs, but the $PU_{1}$ is discarded for not having a large enough bandwidth to be considered a transmission.

**Figure 11 sensors-23-05209-f011:**
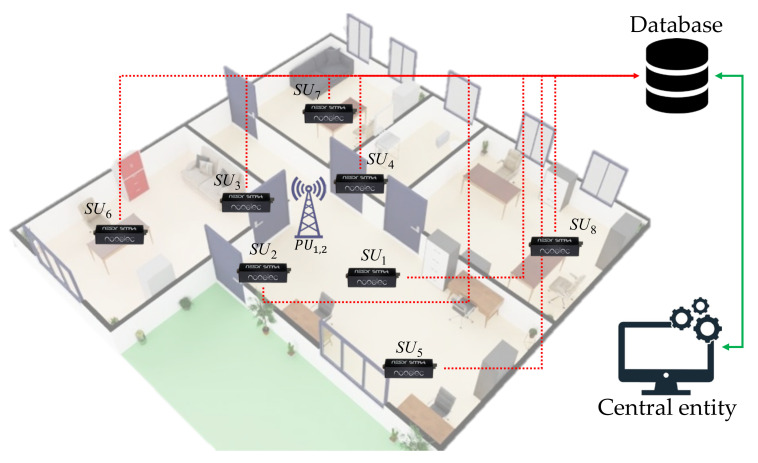
The real implemented scenario.

**Figure 12 sensors-23-05209-f012:**
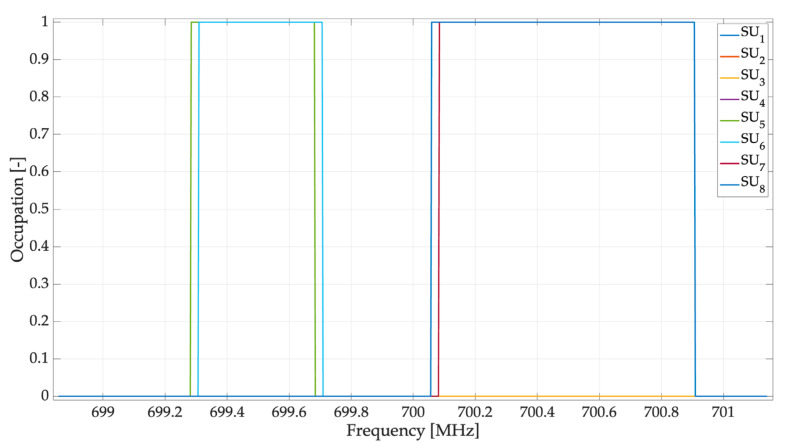
Spectrum occupancy for each SU.

**Figure 13 sensors-23-05209-f013:**
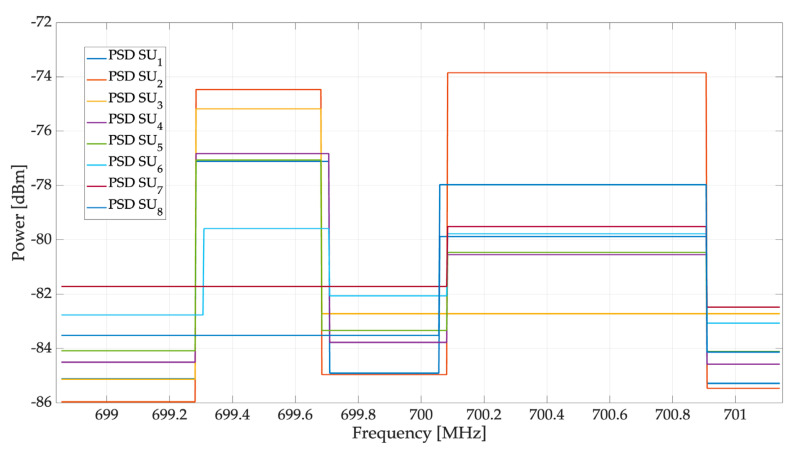
Approximated PSD for each SU.

**Figure 14 sensors-23-05209-f014:**
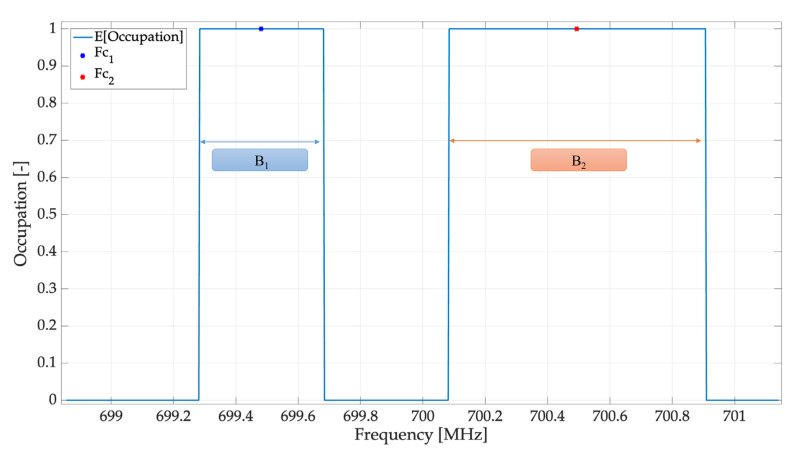
Average occupancy obtained by the central entity.

**Figure 15 sensors-23-05209-f015:**
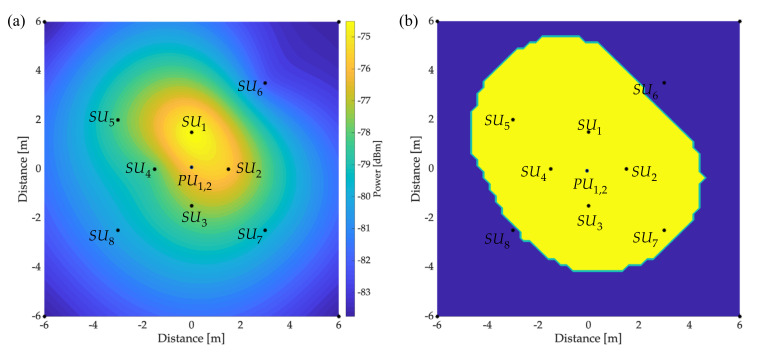
Primary transmission in the 699.48 MHz band with a B of 0.4 MHz: (**a**) REM and (**b**) occupied area.

**Figure 16 sensors-23-05209-f016:**
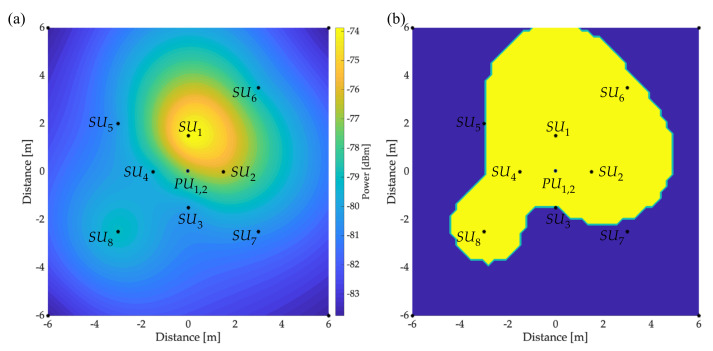
Primary transmission in the 700.49 MHz band with a B of 0.825 MHz: (**a**) REM and (**b**) occupied area.

**Figure 17 sensors-23-05209-f017:**
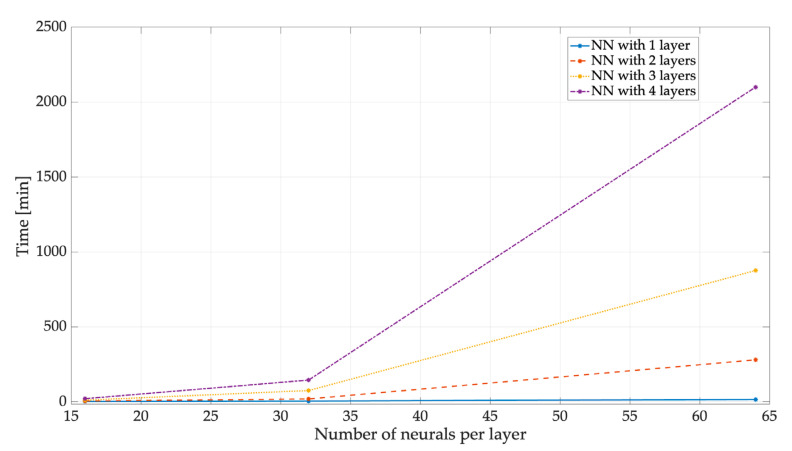
The training time for the different versions of the $NN_{3}$.

**Figure 18 sensors-23-05209-f018:**
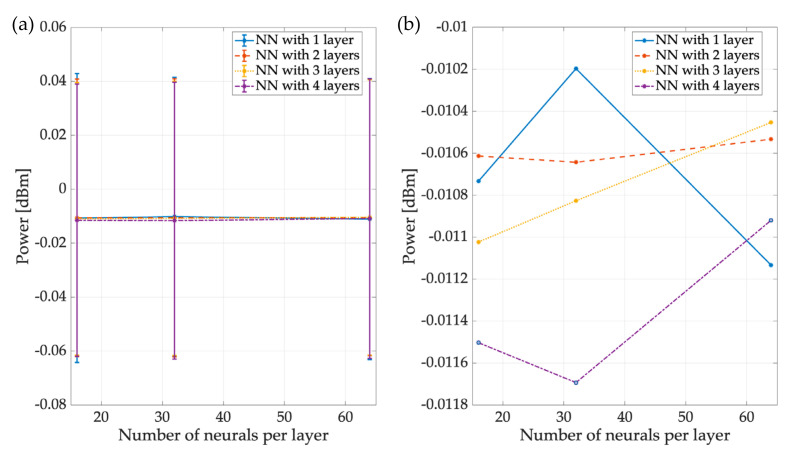
Difference between real values and those obtained by $NN_{1}$: (**a**) mean and standard deviation (STD) and (**b**) mean.

**Figure 19 sensors-23-05209-f019:**
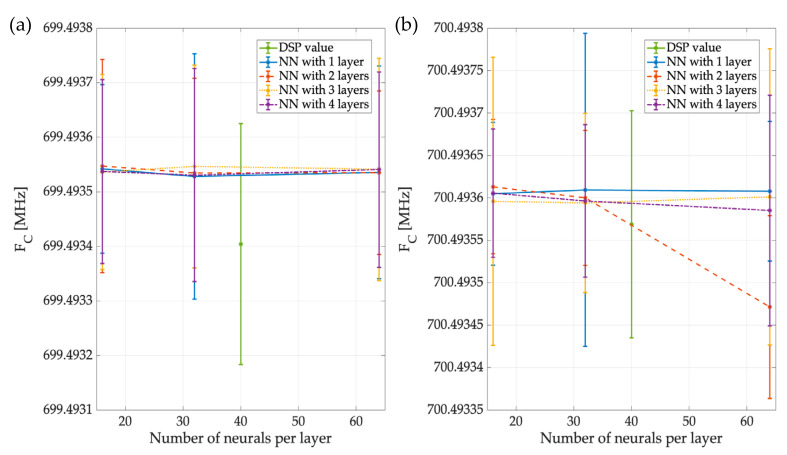
Mean and STD of the precision in detecting (**a**) the $PU_{1}$ carrier of the different $NN_{2}$ variants and (**b**) the $PU_{2}$ carrier of the different networks.

**Figure 20 sensors-23-05209-f020:**
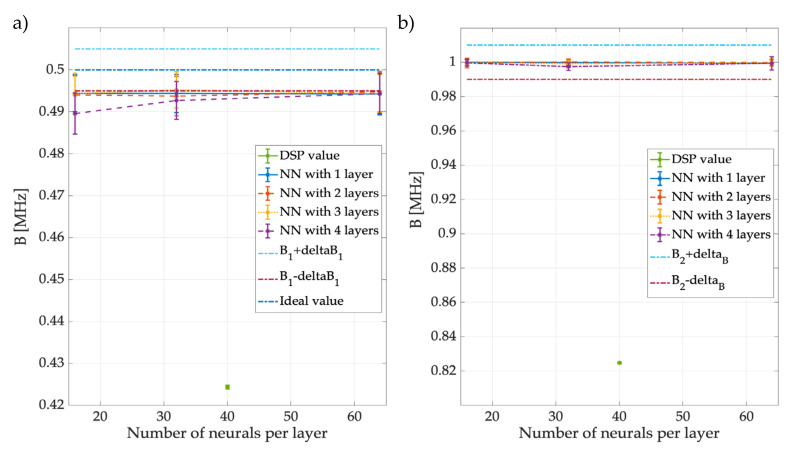
Mean and STD of the precision in detecting (**a**) the $B_{1}$ of different $NN_{3}$ variants and (**b**) the $B_{2}$ of the different networks.

**Figure 21 sensors-23-05209-f021:**
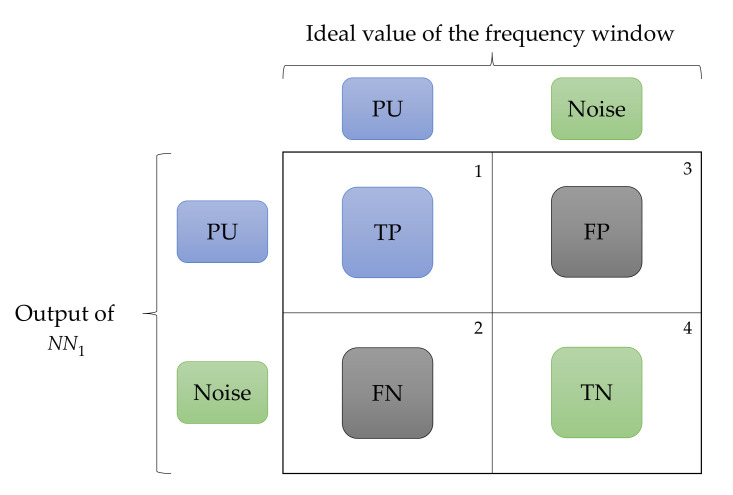
Evaluation outputs of detected windows.

**Figure 22 sensors-23-05209-f022:**
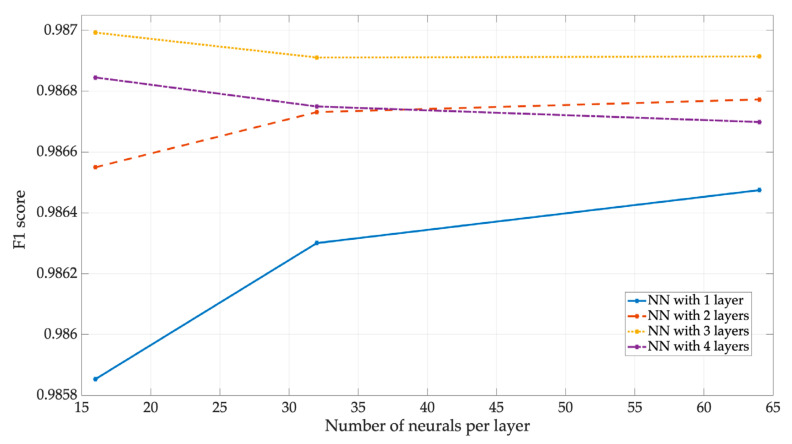
*F*1 score for *NN*_1_.

**Figure 23 sensors-23-05209-f023:**
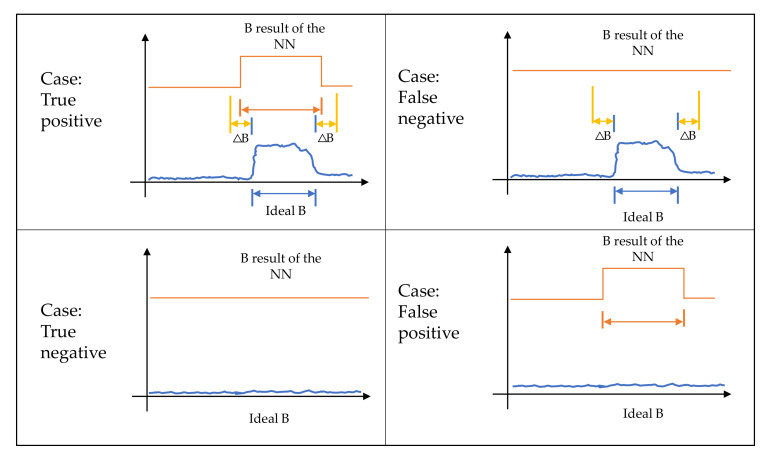
B assessment results.

**Figure 24 sensors-23-05209-f024:**
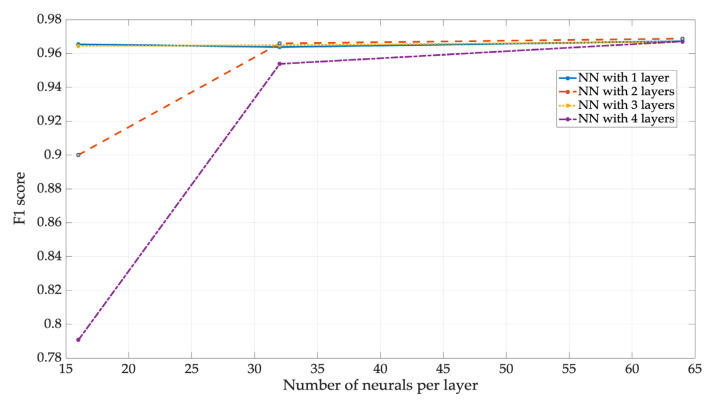
F1 of $NN_{3}$.

**Figure 25 sensors-23-05209-f025:**
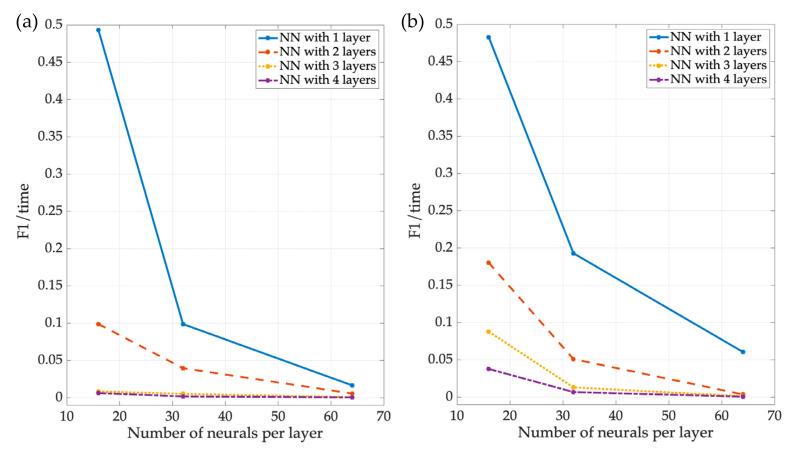
F1 and training time ratio of (**a**) $NN_{1}$ and (**b**) $NN_{3}$.

**Figure 26 sensors-23-05209-f026:**
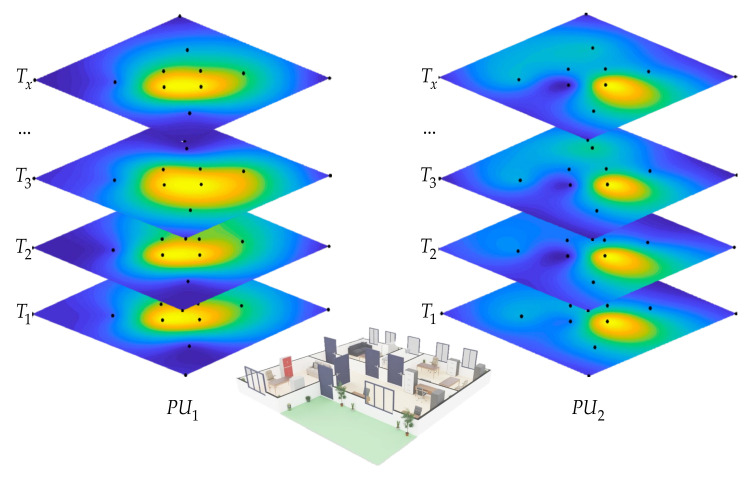
Result of the central entity.

**Table 1 sensors-23-05209-t001:** Main contributions of previous works.

Previous Work.	Contributions
“A Novel Multiband Spectrum Sensing Method Based on Wavelets and the Higuchi Fractal Dimension” [9].	- Two MBSS techniques: Wavelets and multiresolution analysis (MRA). Decision rule based on the Higuchi fractal dimension (HFD). - 95% effective in detecting PUs for SNR values higher than 0 dB. - Controlled simulated environment. - Blind technique.
“Multiband Spectrum Sensing Based on the Sample Entropy” [40].	- MBSS technique based on SampEn. - Technique implemented in a real environment of wireless communications. - Technique with real-time operation (updated every 100 ms). - Probability of 0.99 to correctly detect the PU for SNR values greater than 0 dB. - Cooperation of the different SUs to sense a wide range of frequency. - Blind technique.
“Real-Time Implementation of Multiband Spectrum Sensing Using SDR Technology” [11].	- MBSS technique based on MRA, ML, and HFD implemented in a real wireless communications environment. - Real-time operation (update every 100 ms). - Implemented with SDR devices. - A module is proposed for the elimination through the detailed coefficients obtained with the MRA. - Efficiency of 95% for SNR values greater than 0 dB. - Blind technique.
“Machine Learning Techniques Applied to Multiband Spectrum Sensing in Cognitive Radios” [41].	- Three machine learning methods are applied to the MBSS - Effectiveness of 98% in detecting PUs in the spectrum for SNR values greater than 0 dB. - Controlled simulated environment. - Blind technique.

**Table 2 sensors-23-05209-t002:** Information that each SU shares in the database at each sensing time $T_{x}$.

	$T_{1}$	$T_{2}$	…	$T_{x}$
$SU_{1}$	$\begin{array}{l} {\left[L_{1,1},L_{1,2},\ldots ,L_{1,N}\right]}_{T_{1}}\\ {\left[b_{1,1},b_{1,2},\ldots ,b_{1,N-1}\right]}_{T_{1}}\\ {\left[P_{1,1},P_{1,2},\ldots ,P_{1,N-1}\right]}_{T_{1}} \end{array}$	$\begin{array}{l} {\left[L_{1,1},L_{1,2},\ldots ,L_{1,N}\right]}_{T_{2}}\\ {\left[b_{1,1},b_{1,2},\ldots ,b_{1,N-1}\right]}_{T_{2}}\\ {\left[P_{1,1},P_{1,2},\ldots ,P_{1,N-1}\right]}_{T_{2}} \end{array}$	…	$\begin{array}{l} {\left[L_{1,1},L_{1,2},\ldots ,L_{1,N}\right]}_{T_{x}}\\ {\left[b_{1,1},b_{1,2},\ldots ,b_{1,N-1}\right]}_{T_{x}}\\ {\left[P_{1,1},P_{1,2},\ldots ,P_{1,N-1}\right]}_{T_{x}} \end{array}$
$SU_{2}$	$\begin{array}{l} {\left[L_{2,1},L_{2,2},\ldots ,L_{2,N}\right]}_{T_{1}}\\ {\left[b_{2,1},b_{2,2},\ldots ,b_{2,N-1}\right]}_{T_{1}}\\ {\left[P_{2,1},P_{2,2},\ldots ,P_{2,N-1}\right]}_{T_{1}} \end{array}$	$\begin{array}{l} {\left[L_{2,1},L_{2,2},\ldots ,L_{2,N}\right]}_{T_{2}}\\ {\left[b_{2,1},b_{2,2},\ldots ,b_{2,N-1}\right]}_{T_{2}}\\ {\left[P_{2,1},P_{2,2},\ldots ,P_{2,N-1}\right]}_{T_{2}} \end{array}$	…	$\begin{array}{l} {\left[L_{2,1},L_{2,2},\ldots ,L_{2,N}\right]}_{T_{x}}\\ {\left[b_{2,1},b_{2,2},\ldots ,b_{2,N-1}\right]}_{T_{x}}\\ {\left[P_{2,1},P_{2,2},\ldots ,P_{2,N-1}\right]}_{T_{x}} \end{array}$
…	…	…	…	…
$SU_{8}$	$\begin{array}{l} {\left[L_{8,1},L_{8,2},\ldots ,L_{8,N}\right]}_{T_{1}}\\ {\left[b_{8,1},b_{8,2},\ldots ,b_{8,N-1}\right]}_{T_{1}}\\ {\left[P_{8,1},P_{8,2},\ldots ,P_{8,N-1}\right]}_{T_{1}} \end{array}$	$\begin{array}{l} {\left[L_{8,1},L_{8,2},\ldots ,L_{8,N}\right]}_{T_{2}}\\ {\left[b_{8,1},b_{8,2},\ldots ,b_{8,N-1}\right]}_{T_{2}}\\ {\left[P_{8,1},P_{8,2},\ldots ,P_{8,N-1}\right]}_{T_{2}} \end{array}$	…	$\begin{array}{l} {\left[L_{8,1},L_{8,2},\ldots ,L_{8,N}\right]}_{T_{x}}\\ {\left[b_{8,1},b_{8,2},\ldots ,b_{8,N-1}\right]}_{T_{x}}\\ {\left[P_{8,1},P_{8,2},\ldots ,P_{8,N-1}\right]}_{T_{x}} \end{array}$

**Table 3 sensors-23-05209-t003:** PU and SU main parameters.

Label	Device	$F_{C}\;\mathrm{Tx}\;(\mathrm{MHz})$	$F_{C}\;\mathrm{Rx}\;(\mathrm{MHz})$	Bandwidth (MHz)	Location Coordinate (X,Y) (m)
*PU_1_*	Mini LimeSDR	699.5	-	0.5	(0, 0)
*PU_2_*	HackRF ONE	700.5	-	1	(0, 0)
*SU_1_*	RTL-SDR	-	700	2.4	(−1.5, 0)
*SU_2_*	RTL-SDR	-	700	2.4	(0, 1.5)
*SU_3_*	RTL-SDR	-	700	2.4	(1.5, 0)
*SU_4_*	RTL-SDR	-	700	2.4	(0, −1.5)
*SU_5_*	RTL-SDR	-	700	2.4	(−3, 2)
*SU_6_*	RTL-SDR	-	700	2.4	(3, 3.5)
*SU_7_*	RTL-SDR	-	700	2.4	(3, −2.5)
*SU_8_*	RTL-SDR	-	700	2.4	(−3, −2.5)

**Table 4 sensors-23-05209-t004:** The training time of (a) $NN_{1}$, (b) $NN_{2}$, and (c) $NN_{3}$.

$\mathrm{Training}\;\mathrm{Time}\;NN_{1}\;[\min]$				
(a)		Neurons		
		16	32	64
Layers	1	2	10	60
	2	10	25	180
	3	120	192	1500
	4	162	600	3984
Training Time $NN_{2}$ [min]				
(b)		Neurons		
		16	32	64
Layers	1	2	6	16
	2	8	33	290
	3	15	80	850
	4	20	129	7610
Training Time $NN_{3}$ [min]				
(c)		Neurons		
		16	32	64
Layers	1	2	5	16
	2	5	19	280
	3	11	75	875
	4	21	145	3628

## Data Availability

Data sharing not applicable.

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
