# Peer review of "Cooperative Multiband Spectrum Sensing Using Radio Environment Maps and Neural Networks"

_sensors, 2023, doi:10.3390/s23115209_

Round 1

Reviewer 1 Report

This paper studies cooperative spectral sensing which is a vital part of the cognitive radio networks (CRN) by employing radio environment maps (REM) and neural works. The topic is timely since CRN is the promising network to improve the spectral efficiency of wireless networks and is suitable for publication in Sensors as it collects results from lots of sensors. There are some weak points that are given below

The results mainly employ in the pre-defined areas where the number of primary users as well as their characteristics (transmit power, locations, bandwidth, etc.) are given in advance. Nonetheless, in practical scenarios, it is hard to assume that information is available and correct. Thus, the reviewer invites the authors to discuss the applications in real environments with unknown information.

There are many works in the literature that study spectrum sensing; hence, it is mandatory to compare the proposed solution with works in the literature.

It is recommended to summarize the hyperparameters of the neural networks so that readers are easy to follow.

Some equations not proposed by the authors should be cited in the original work

The presentation is okay.

Reviewer 2 Report

The paper presents solid work in the area of cooperative spectrum sensing. The strength of the paper is that the authors report the use of neural networks in a real wireless communications scenario to perform cooperative multiband spectrum sensing, thus showing that the proposal can be implemented in real-time,  which is quite interesting. However, some minor corrections could be done prior to its publication that might help to improve the quality of the paper. 

Regarding the presentation of the paper, the quality of all the figures must be improved since they cannot be seen clearly when the paper is printed. Please increase the font size used in all the figures, also, increase the font size of the x and y labels. Specifically, in figures 12, 13, 14, 17, 18, 19, 20, 22, 23, 24, and 25, increase the line width, marker size, font size of legends, and font size of the x and y labels.  

Please use punctuation marks such as commas and full stops in all the mathematical equations as necessary. 

Finally, according to the review I performed, I think the paper shows sufficient merits to be published as a journal paper.

Reviewer 3 Report

The authors have reported a Cooperative multiband spectrum sensing using radio environment maps and neural networks. I have a few suggestions that should be considered in the manuscript:

1.     Fig. 22, 24 and 25, the y-axis parameters are not clear. Please correct these figures.

2.     Please add the experimental results if possible.

3.     Add a comparison table to justify the novelty of your work to be published past years.

4.     The conclusion part is so large. Please write the constructive word.

The overall quality of the paper is good, and I suggest minor revisions for this paper.

Good

Reviewer 4 Report

The contributions of this paper include the proposal of a centralized network of cognitive radios for monitoring a multiband spectrum in real-time, the use of a monitoring technique based on sample entropy to determine secondary users' spectrum occupancy, the uploading of determined features of potentially detected primary users to a database, and the estimation of the number of primary users, their carrier frequency, bandwidth, and the spectral gaps in a given area through the monitoring technique.

The findings of the proposed methodology's implementation were conducted in a real wireless communication environment. The results are broken down into two subsections: the central entity based on digital signal processing is covered in the first, and the central entity based on neural networks is covered in the second. Both suggested cognitive networks provide timely information to secondary users for transmission, effectively diagnose main users, and prevent concealed terminal issues. The cognitive radio network using neural networks, however, performs better at identifying the carrier frequency and bandwidth of major users.

The following are some of this paper's limitations:

The proposed cognitive radio network is evaluated in a controlled setting, and the outcomes may differ in actual usage.

The suggested technique for locating primary users makes the assumption that these users are stationary and that their features don't change over time.

Large-scale networks might not be able to support the proposed cognitive radio network's requirement for a centralized organization to process the submitted data.

The proposed cognitive radio network does not take into account how secondary user interference affects the ability to recognize primary users.

The impact of fading and multipath propagation on the identification of principal users is not taken into account by the proposed cognitive radio network.

Figure 12 and 13: the lines are not clear. Modification is highly recommended.

The conclusion may be shorter and more compact. 

Round 2

Reviewer 4 Report

The authors answered all the raised questions